# Assessing the Feasibility of a Multimodal Approach to Pain Evaluation in Early Stages after Spinal Cord Injury

**DOI:** 10.3390/ijms241311122

**Published:** 2023-07-05

**Authors:** Simona Capossela, Gunther Landmann, Mario Ernst, Lenka Stockinger, Jivko Stoyanov

**Affiliations:** 1Swiss Paraplegic Research, CH-6207 Nottwil, Switzerland; simona.capossela@paraplegie.ch (S.C.); mario.ernst@paraplegie.ch (M.E.); 2Centre for Pain Medicine, Swiss Paraplegic Centre, CH-6207 Nottwil, Switzerland; gunther.landmann@paraplegie.ch (G.L.); lenka.stockinger@paraplegie.ch (L.S.); 3Faculty of Health Sciences and Medicine, University of Lucerne, CH-6002 Lucerne, Switzerland; 4Institute of Social and Preventive Medicine, University of Bern, CH-3012 Bern, Switzerland

**Keywords:** spinal cord injury, neuropathic pain, quantitative sensory testing, molecular biomarkers

## Abstract

This research evaluates the feasibility of a multimodal pain assessment protocol during rehabilitation following spinal cord injury (SCI). The protocol amalgamates clinical workup (CW), quantitative sensory testing (QST), and psychosocial factors (PSF) administered at 4 (T1), 12 (T2), and 24 (T3) weeks post injury and at discharge (T4). Molecular blood biomarkers (BB) were evaluated via gene expression and proteomic assays at T1 and T4. Different pain trajectories and temporal changes were identified using QST, with inflammation and pain-related biomarkers recorded. Higher concentrations of osteopontin and cystatin-C were found in SCI patients compared to healthy controls, indicating their potential as biomarkers. We observed altered inflammatory responses and a slight increase in ICAM-1 and CCL3 were noted, pointing towards changes in cellular adhesion linked with spinal injury and a possible connection with neuropathic pain. Despite a small patient sample hindering the correlation of feasibility data, descriptive statistical analyses were conducted on stress, depression, anxiety, quality of life, and pain interferences. The SCI Pain Instrument (SCIPI) was efficient in distinguishing between nociceptive and neuropathic pain, showing a progressive increase in severity over time. The findings emphasize the need for the careful consideration of recruitment setting and protocol adjustments to enhance the feasibility of multimodal pain evaluation studies post SCI. They also shed light on potential early adaptive mechanisms in SCI pathophysiology, warranting the further exploration of prognostic and preventive strategies for chronic pain in the SCI population.

## 1. Introduction

Pain is the most reported problem in patients with spinal cord injury (SCI) [1]. In Switzerland, 74% of SCI patients have chronic pain, prevalently musculoskeletal (71%) and neuropathic (62%) [2]. Musculoskeletal pain occurs either early after injury and slightly decreases within the first year or evolves in the following years [3,4]. The development of neuropathic pain early after injury is mostly of unclear etiology, difficult to treat [4,5], and likely to predict chronicity [3]. A meta-analysis on neuropathic pain found that 72% of participants had pain 6 months after injury [6]. There is a scarcity of longitudinal studies examining pain onset, following trajectories of pain (changes over time), and identifying factors that determine these trajectories.

In SCI patients with chronic pain, the pain intensity is correlated with higher depressive symptoms and lower quality of life [7]. Positive psychology interventions as part of chronic pain treatment have beneficial effects on pain intensity and emotional functioning [8,9,10].

As pain development and perception has multiple bio-psychosocial components, it would make sense to study as many of these components as possible and identify pain trajectories at an early stage after SCI, in order to define an appropriate pain intervention [11].

Quantitative sensory testing (QST) is an established diagnostic tool to monitor sensory profiles in chronic neuropathic pain [12]. QST has been performed at early stages of neuropathic SCI pain (SCIP) [4,13]. The QST protocol of the German Research Network on Neuropathic Pain [14] is widely used in investigations of neuropathic pain syndromes [12]. However, studies in SCIP using this protocol have only been performed in cases of chronic SCI [15,16].

Lately, blood biomarker (BB) analysis has gradually become part of the studies of chronic pain. Changes in inflammatory mediators in long-term medically treated patients with persistent chronic pain are known; however, sometimes findings may be a result of long courses of anti-inflammatory therapy [17]. On the contrary, longitudinal descriptions of pro- and anti-inflammatory marker levels, starting shortly after the SCI and time-related association with the development of neuropathic pain, are not available.

The aim of this study was to evaluate the feasibility of a multimodal pain evaluation protocol during the first rehabilitation after SCI. The protocol included an evaluation of neurological pain with clinical workup (CW) and QST to identify longitudinal patterns (trajectories) of neuropathic pain in individuals with newly diagnosed SCI. The analysis of blood biomarkers (BBs) and psychosocial factors (PSFs) at consecutive timepoints during the first rehabilitation after SCI was also included in the protocol, in order to determine predictors and identify potential biomarkers for the development of neuropathic pain.

### Study Population

Our study, which ran from September 2017 to December 2018, initially screened 117 spinal cord injury (SCI) patients. Of these, 53 agreed to participate in the SwiSCI cohort study [18]. From this group, 37 were considered eligible and invited to partake in the SwiSCI nested project. Seven of these individuals underwent a neurological and pain clinical workup/quantitative sensory testing (CW/QST) evaluation. Of the original group, eight consented, but one patient was excluded due to reliance on a respiratory machine. Thirteen participants consented to blood donation for biomarker analysis, five of whom also took part in the neurological and pain evaluation; one patient was excluded from neurological and pain evaluation due to respiratory concerns; and seven patients declined the neurological and pain assessment but consented to blood donation. Fourteen participants completed the SwiSCI questionnaire and underwent PSF assessment. Only one patient who had consented to the neurological and pain evaluation and blood donation could not complete the SwiSCI questionnaire. Further details are provided in Figure 1 and Appendix A.

The SwiSCI nested project aimed to identify longitudinal patterns (trajectories) and ascertain predictors for the development of neuropathic pain. The participants underwent neurological and pain evaluation as well as PSF assessments at 4 (T1), 12 (T2), and 24 (T3) weeks post SCI, and at the conclusion of their initial rehabilitation upon hospital discharge (T4). Blood biomarkers were analyzed at T1 and T4. In our sample (*n* = 15), the end of the initial rehabilitation period (T4) was on average 27 ± 8.8 weeks post injury (95% CI: 22–31).

## 2. Results

### 2.1. Neurological and Pain-Related Assessments

Due to varying lengths of stay in the intensive care unit and different commencement times of regular rehabilitation, several QST data points were unavailable at T1. Additionally, given the flexible nature of the initial rehabilitation completion, T4 coincided with T3 for six patients who underwent QST evaluation. Consequently, for pain trajectory analysis, QST data from T2 and T3 were compared. Data from T1 (N = 2) and T4 (N = 1) are presented in Appendix A.

Figure 2 showcases the pain drawings and pain trajectories. Demographic and additional clinical data can be found in Appendix A.

The total examination time spanned three hours, with one hour dedicated to the neurological visit and two hours to QST. For the initial patient, neurophysiology additionally included laser-evoked potentials (LEP). However, due to the time and resource demands associated with this investigation (an additional hour of examination time and the need for an extra technician), LEP was omitted for all subsequent patients, following an amendment approved by the ethics committee.

The pain interference questionnaire (Table 1) revealed a trend toward increased interference of pain with activity, mood, and sleep from T1 to T2. However, there was an improvement observed from T2 to T4 across all items.

The Spinal Cord Injury Pain Instrument (SCIPI) questionnaires (Table 1) showed an escalation in the severity of neuropathic and nociceptive pain from T1 to T4. One patient experienced the required points for neuropathic pain at T2 and T4, but not at T1.

### 2.2. Psychosocial Factors

PSF were analyzed in the SwiSCI questionnaire (N = 14). Using a descriptive statistical analysis, SCI patients showed 20% lower stress, 8% lower depression, and a 17% increase in anxiety at T4 compared to T1, while there were no differences in QoL score (Table 1). The questionnaires were not always completed by the patients (Appendix A).

### 2.3. Pain Trajectories

Between timepoints T2 and T3, six pain trajectories were identified (Figure 2): pain-free trajectory (pain-free at both timepoints), N = 1; nociceptive pain evolving trajectory (no nociceptive pain at T2, development of nociceptive pain to T3), N = 4; nociceptive pain persistent trajectory (nociceptive pain at both timepoints), N = 1; neuropathic pain evolving trajectory (no neuropathic pain at T2, but new onset of neuropathic pain at T3), N = 1; neuropathic pain persistent trajectory (neuropathic pain present on both timepoints), N = 2; and neuropathic pain subtype change trajectory (change from neuropathic subtype at-level SCIP to below-level SCIP due to change of NLI from C3 to C1), N = 2. Considering T1 and T4 (Appendix A), two more pain trajectories—pain type change trajectory (nociceptive pain at T1, neuropathic pain at T2), N = 1, and nociceptive pain resolving trajectory (nociceptive pain at one timepoint, resolved at following timepoint), N = 2—and another neuropathic pain subtype change trajectory, due to a change in the extent of the neuropathic pain area (whole hand to thumb) according to the subtype definition [19], were identified.

For the identification of pain trajectories, no feasibility limitations were observed except one, where despite clinical–neurological judgement for neuropathic pain, the SCIPI questionnaire did not reach the required amount of points (Table 1).

### 2.4. Quantitative Sensory Testing

QST results, grouped for pain trajectories, are shown in Figure 3. Conclusions cannot be made for three reasons: low patient count enrolled, variations in patient count according to pain trajectory group, and limitations of QST performance. In the neuropathic pain group (N = 2) at T2, above-level QST showed abnormalities in the loss and gain of function in thermal and mechanical parameters (N = 9). At T3, the number of abnormal parameters diminished (N = 4), reflecting some improvement between timepoints. At-level QST showed more abnormalities at T2 (N = 11), and improvement in a lower number of parameters (N = 5) at T3. Several QST parameters could not be determined due to lack of stimuli perception, anatomical reasons, and the technical limitations of the temperature testing device. An overview of the missing QST values and reasons is given in Appendix A. To handle these limitations, ongoing adaptations of SOP were made and further test sites in addition to those in the literature [14,20] were defined (Appendix A). Finally, impaired hand function in tetraplegia interferes with the handling of a computer mouse during QST. A large-button (7.5 cm diameter) mouse, which could be pressed with the whole dorsum of the hand or forearm in cases of tetraplegia-related finger weakness, was used. Patients with additional weakness of the arm or forearm need to be checked in order to determine whether this sort of button might be feasible.

### 2.5. Blood Biomarkers

The samples of SCI patients who had consented to donate blood (N = 13; 12 males, 1 female, age 47.7 ± 17.6 years) were assayed for inflammation and pain-related BB at T1 and T4 and compared to six healthy able-bodied individuals (CTR; males, age 36 ± 12 years). Demographic and clinical data are provided in Appendix A.

Gene expression was feasible for all 14 genes. We found lower levels of MCP-1 and interleukin (IL)-6 in SCI compared to CTR and changes in IL-10 and CCL3 between T1 and T4 (Figure 4). At the protein level, BBs were analyzed in serum by immunoassay. Among 26 proteins, 6 were not or were barely detectable. We found a decrease in C-reactive protein (CRP), calprotectin, and osteopontin (OPN) and a slight increase in ICAM-1 at T4 compared to T1. Excluding the healthy donor with a high CRP level (12.9 mg/L), the CRP value was significantly (*p*= 0.04) higher in SCI patients at timepoint T1 compared to healthy controls (2.4 ± 0.8 mg/L, *n* = 3). Higher concentrations of OPN and cystatin-C were found in SCI patients, compared to CTR (Figure 5). No differences were found for the other blood biomarkers (Appendix A). Considering the donors who consented to both neurological pain evaluation and blood donations (*n* = 5), significance was reached only for MCP1, IL6, CCL3, calprotectin, OPN (CTR vs. T1, CTR vs. T4) and cystatin C (CTR vs. T4).

### 2.6. Feasibility Overview and Feedback

The study organization of a multimodal pain assessment protocol was focused on feasibility. We could manage QST and pain workup at T2 and T3, define pain trajectories, measure blood biomarkers, and collect PSF evaluation within SwiSCI questionnaire. However, the recruitment rate was very low, QST and pain workup were not possible at T1 and T4 for all samples, the combination of QST and LEP was too long, the PSF questionnaire was not always filled in for all fields, and only a descriptive statistical analysis was carried out.

The evaluation of anonymized QST feedback questionnaires is shown in Appendix A. The feedback reached highest agreement. Most of the patients rated the measurement duration as adequate. No negative feedback was given, and all patients recommended participation to equally affected individuals. Internal feedback by study staff revealed minor procedural problems related to difficulties in appointment booking. Finally, these findings did not interfere with the study proceedings at all.

## 3. Discussion

This study evaluated the feasibility of characterizing different pain trajectories using a multimodal pain evaluation protocol that combines clinical workup (CW), quantitative sensory testing (QST), blood biomarker (BB), and psychosocial factor (PSF) assessments at various timepoints, during initial post spinal cord injury (SCI) rehabilitation. Four main challenges were identified which will need to be considered in future study designs: a lower than expected recruitment rate, the burden of multiple assessments, hurdles in QST evaluation, and the collection of self-reported data.

Although no major deviations from the study protocol occurred post inclusion, the unexpectedly low recruitment rate poses a challenge for comprehensive data acquisition. In the study proposal was envisioned a recruitment of 150 SCI patients, but out of 117 SCI patients, only 31% consented to take part inthe SwiSCI cohort study, and only 22% of them (8/37) consented to participate to the neurological pain evaluation within the SwiSCI nested project. This was below the level of recruitment reported in studies involving QST [4,13,15,21,22,23], except for a study which reported a recruitment rate of 1.2% (12/998) in an outpatient setting [16]. In hindsight, a better time to start the study should have been selected, as other studies were recruiting at the same time in our hospital, leading to ‘‘consent fatigue’’. Furthermore, the 4 h initial duration of the assessments was given as a reason for consent refusal by patients. We adapted by shortening the study time by one hour (omitting LEP). The combined results of QST and LEP in our first patient have been published [24], showing the potential feasibility of this combination for patients with acute SCI.

The importance of early examinations is emphasized by the discovery of new pain trajectories shortly after injury. The identified trajectories may suggest changes in pain pathways in the early stage of SCI, highlighting the potential for early intervention strategies. QST evaluation was not always feasible at T1 due to conflict with rehabilitation scheduling. Similarly, recruitment limitations due to surgery or stays in intensive care units in studies involving QST in SCI have been reported [4,13].

The clinical definition of a total of eight types of pain trajectory was feasible. Pain-free, development of nociceptive or neuropathic pain, persistent nociceptive or neuropathic pain, and the resolution of nociceptive pain have been reported [3,4,13]. Further pain types such as the change from nociceptive to neuropathic pain and the change in neuropathic pain subtype are discussed in our case report [24] and previous studies [25].

Performing QST in patients with SCI revealed that several adaptations are needed in the protocols [14,20]. Several parameters could not be tested because stimuli were not felt, due to partial or complete neuronal deafferentation. QST in complete SCI will show a deafferentation QST, which cannot be interpreted properly. Previous studies have avoided this problem by choosing examination sites only in the incomplete lesion area or in complete lesions within the partial preservation zone [13,15,16,21]. Furthermore, it was considered that patients with complete SCI will not have sensory hypersensitivity below the injury [4]. In patients that experienced difficulties using a computer mouse due to lesion-associated hand weakness, we used a large button, which could be pressed with the upper arm or forearm. This adaptation has not yet been described. Most QST studies in SCI omitted patients with a lesion above T1 [16], a complete lesion involving the arm area and other feasibility issues in performing QST [15].

In our study, 35% of SCI patients eligible to participate in the SwiSCI nested project (13/37) consented to donate blood, a rate below that of the routine SwiSCI biobank participation rate [26]. The analysis of blood biomarkers was successfully conducted for all 13 patients via gene expression and proteomic assays, revealing promising molecules that may serve as future therapeutic targets or diagnostic tools. Notably, IL-10, CCL3, MCP1, IL-6, CRP, calprotectin, cystatin-C, OPN, and ICAM-1 exhibited significant alterations. These molecules are known to play crucial roles in inflammatory responses, spinal cord injury, and pain modulation, which could have a direct impact on therapeutic strategies for SCI-related pain. Gene expression analysis showed higher levels of the anti-inflammatory IL-10 at T1, which has been demonstrated to have potential as a treatment for persistent inflammatory pain [27], and at T4 a slight increase in CCL3, highly associated with SCI, induced chronic neuropathic pain in a mouse model [28]. Lower levels of MCP1 and IL-6 were found in SCI patients compared to healthy controls, suggesting an altered inflammatory response in the patient group. Increased serum levels of pro-inflammatory cytokines have been previously correlated with increased pain intensity [29] but a decrease in MCP1 was found in medically treated patients with chronic pain [17]. The interpretation of biomarker values in the context of therapy in the acute stage of SCI remains open. Patients were treated with anti-thrombotic, anti-depressive, non-opioid, and opioid analgesic (Appendix A). At the protein level, study participants showed higher levels of CRP and calprotectin at T1, with a decrease at T4 to normal levels. Elevated CRP levels have been described in chronic SCI [30] and calprotectin could also be representative of systemic inflammation [31]. Furthermore, SCI patients showed higher levels of cystatin-C and OPN compared to healthy controls, indicating their potential as biomarkers. In addition, OPN was higher at T1 compared to T4. Increased serum cystatin-C was found in SCI patients, with higher levels in the more severely injured individuals [32]. OPN is routinely expressed by injured tissues [33] and has a potential neuroprotective role in the inflammatory response to spinal cord injury. We also found at T4 a slight increase in ICAM-1, which has been reported in SCI patients with pressure ulcers [34], pointing towards changes in cellular adhesion linked with spinal injury.

Moreover, our research’s practical implications should not be overlooked. Specifically, the five patients who underwent the entire protocol provide key insights into the therapeutic prospects that arise from such a comprehensive approach. This knowledge can guide the development of personalized, integrative care models for SCI-related pain, and potentially improve patient outcomes. Due to the low number of patients (N = 5), who consented to both the feasibility study and biobank blood donation, QST and molecular biomarker data could not be properly correlated.

The analysis of PSF was feasible, even if the questionnaires were not always completed by the patients. Descriptive statistical analysis showed no difference in quality of life but a decrease in stress and depression and increase in anxiety after the first rehabilitation. Small changes in pain interference with daily activities, mood, and sleep were observed, with an increase at T2, followed by a slight decrease at T3. Chronic pain in SCI was associated with reduced social participation, depressive symptoms, and worse quality of life [7], but psychology interventions were shown to reduce pain and improve functioning [8]. Compared to clinical evaluation, in line with validated sensitivity and specificity [35], the SCIPI questionnaire [36] showed symptom progression over time and longitudinal discrimination between nociceptive and neuropathic pain, except in one case of neuropathic pain.

This study faced many obstacles but showed that a multimodal bio-psychosocial pain evaluation protocol was feasible in the early stages of SCI, even though the recruitment setting, time resources, and assessment protocols need to be adjusted in a successful SCI-related pain study.

Importantly, the preliminary results of this feasibility study indicate possible early adaption mechanisms that are worthy of further exploration as candidates for prognostic and preventive strategies in relation to chronic pain in the SCI population.

## 4. Materials and Methods

### 4.1. Study Design

This feasibility study is a nested project within the Swiss Spinal Cord Injury (SwiSCI) inception cohort study [18]. SwiSCI is a prospective observational cohort study that collect demographic, biopsychosocial, clinical parameters, and biological samples from persons who reside in Switzerland and have been diagnosed with traumatic or non-traumatic SCI. Data are collected at 5 measurement time points after SCI: T1 (4 weeks ± 12 days), T2 (12 weeks ± 14 days), T3 (24 weeks ± 18 days), T4 (end of first rehabilitation—discharge from the hospital), and T5 (1 year after discharge) [26]. The SwiSCI nested project is focused on identifying longitudinal patterns (trajectories) and determining predictors for the development of neuropathic pain in the early stages after SCI using three types of evaluation method: clinical workup (CW), quantitative sensory testing (QST), psychosocial factors (PSFs), and molecular blood biomarkers (BBs), described in detail below.

### 4.2. Patient Recruitment

In the SwiSCI nested project study proposal was envisioned a recruitment of 150 SCI patients in 3 years. A total sample of 118 patients were supposed to provide an 80% power to reject the null hypothesis with an α level of 0.05 and an estimated effect size of Cohen’s d of 0.5 (in the case of any pain treatment).

Initially, SCI patients were screened by a clinical trial unit to participate in the SwiSCI study. Those who consented were invited to fill in the SwiSCI questionnaire (including PSF), asked to donate a blood sample (at T1 and T4), and/or participated in the nested project, including neurological and pain CW/QST evaluation. Eligibility criteria were based on the SwiSCI inception cohort standard and included all Swiss residents with new SCI, recruited at the time of entry to their first rehabilitation program in one of the collaborating rehabilitation centers [18]. Exclusion criteria were language and non-traumatic SCI. Later on, due to low recruitment, non-traumatic SCI was no longer considered an exclusion criterion.

### 4.3. Neurological and Pain-Related Assessments

The CW assessments included international standards for neurological classification of spinal cord injury [37] to assess motor and sensory impairment in SCI; International Spinal Cord Injury Pain Classification [19] to diagnose pain types as nociceptive, neuropathic, or other pain, including subtypes; International Spinal Cord Injury Pain Basic Data Set version 2.0 [38], using pain drawings, pain types and subtypes, mean pain intensity, onset of pain, pain treatment, and a pain interference questionnaire, rated 0–10 (no interference–maximum interference), considering daily activities, mood, and sleep; and the 7-item version of the Spinal Cord Injury Pain Instrument (SCIPI) [35], a validated screening tool used to distinguish between neuropathic (score above 3.5) and nociceptive pain.

### 4.4. Psychosocial Factors

In the SwiSCI questionnaire dataset, the following PSFs were obtained: distress thermometer, a visual analogue scale ranging from 0 to 10 (no distress–extreme distress), used to rate emotional distress due to SCI (5 means moderate distress) [39]; the Hospital Anxiety and Depression Scale (HADS), which consists of 14 items with subscale score 0–21 in order to assess levels of anxiety and depression (score > 8) [40,41]; and quality of life (QoL) satisfaction, with 3 variables (general quality of life, physical health, and psychological health) on a scale ranging from 0 to 10 (completely dissatisfied–satisfied) [42].

### 4.5. Quantitative Sensory Testing

QST was performed according to the standardized protocol [14] at 3 different locations on the left body site: above-level area, within 2 dermatomes above the neurological level of injury (NLI) as an unaffected control area; at-level area, within 3 dermatomes below the NLI (suspected to correlate with the area of neuropathic at-level SCIP and associated sensory findings); and below-level area, at the dorsum of the foot.

### 4.6. Blood Biomarkers

RNA and serum were provided by SwiSCI Biobank, Nottwil, Switzerland. Control samples (CTR) were collected from volunteers, healthy able-bodied individuals, after written informed consent. RNA was used for cDNA synthesis (VILO, LubioScience, Zurich, Switzerland) and amplified with specific primers (Appendix A) by quantitative PCR (BioRad, Cressier, Switzerland). Relative quantification was based on the 2^−∆∆Ct^ method and normalized to 3 housekeeping genes. Serum BB was measured in a multi-plex analysis by flow cytometer (CytoFlex Beckman Coulter, Nyon, Switzerland) with LEGENDplex bead-based immunoassays (Biolegend, London, UK). Data were analyzed with LEGENDplex Biolegend data analysis software (Version 2023_02_15).

### 4.7. Feasibility Feedback

The feasibility of the QST standard operating procedure (SOP) was assessed based on planning, procedure, documentation, evaluation-related issues, and patient feedback with a 9-item questionnaire.

### 4.8. Statistics

For in gene expression and immunoassay analysis, thew non-parametric Mann–Whitney–Wilcoxon U test was used for independent variables to compare SCI and CTR, and the non-parametric Wilcoxon signed-rank test was used for related variables to compare T1 and T4. Data analysis was performed with SPSS version 25.0 for Windows (IBM Corporation, Armonk, NY, USA). Values under the limit of detection (LOD) were not included. A descriptive statistical analysis was performed on all other data.

## Figures and Tables

**Figure 1 ijms-24-11122-f001:**
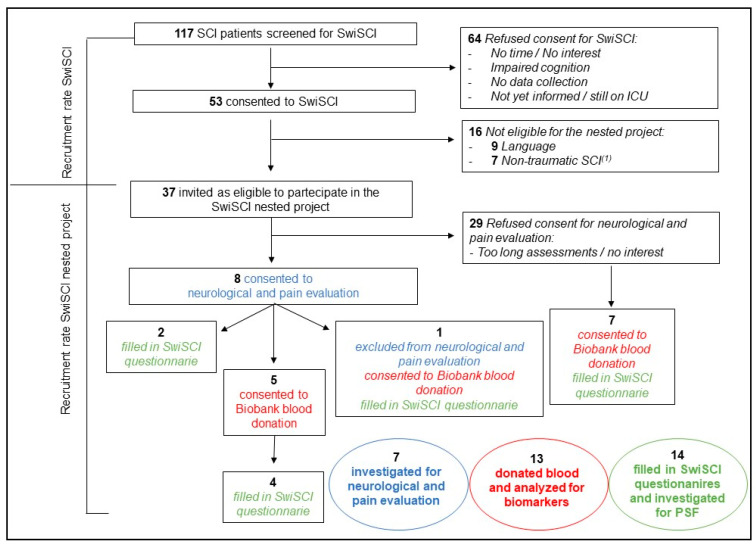
Recruitment flow diagram. This figure illustrates the variability in the number of participants undergoing neurological and pain evaluation, blood biomarker analysis, and psychosocial factor (PSF) assessments. The differing participation levels in each assessment type are due to individual patient decisions to opt out of certain evaluations. The diagram serves as a vital resource for identifying potential challenges in participant recruitment for future multimodal pain studies. **Note:** ^(1)^ Later on, due to low recruitment, non-traumatic SCI was no longer considered an exclusion criteria. **Abbreviations:** ICU: intensive care unit; SCI: spinal cord injury; SwiSCI: Swiss SCI Cohort Study.

**Figure 2 ijms-24-11122-f002:**
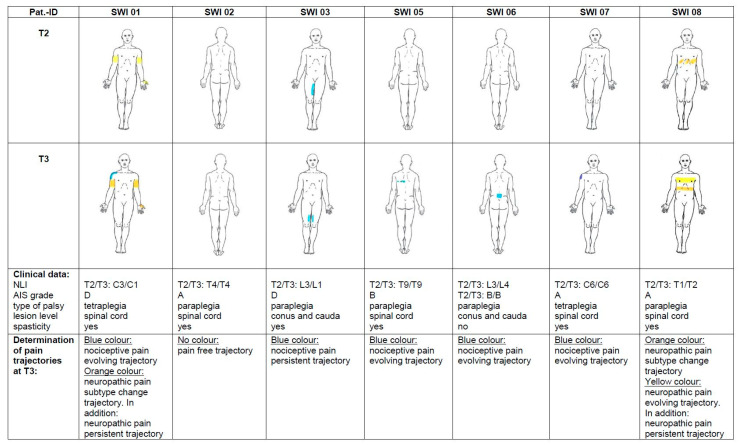
Pain drawings, clinical data, and pain trajectories. Blue color: musculoskeletal pain. Yellow color: neuropathic at-level SCIP. Orange color: neuropathic below-level SCIP. Changes in NLI between timepoints T2 and T3 are indicated. Timepoints: T2 (12 weeks) and T3 (24 weeks). **Abbreviations**: AIS: ASIA (American Spinal Injury Association) Impairment Scale; NLI: neurological level of injury; SCIP: spinal cord injury pain.

**Figure 3 ijms-24-11122-f003:**
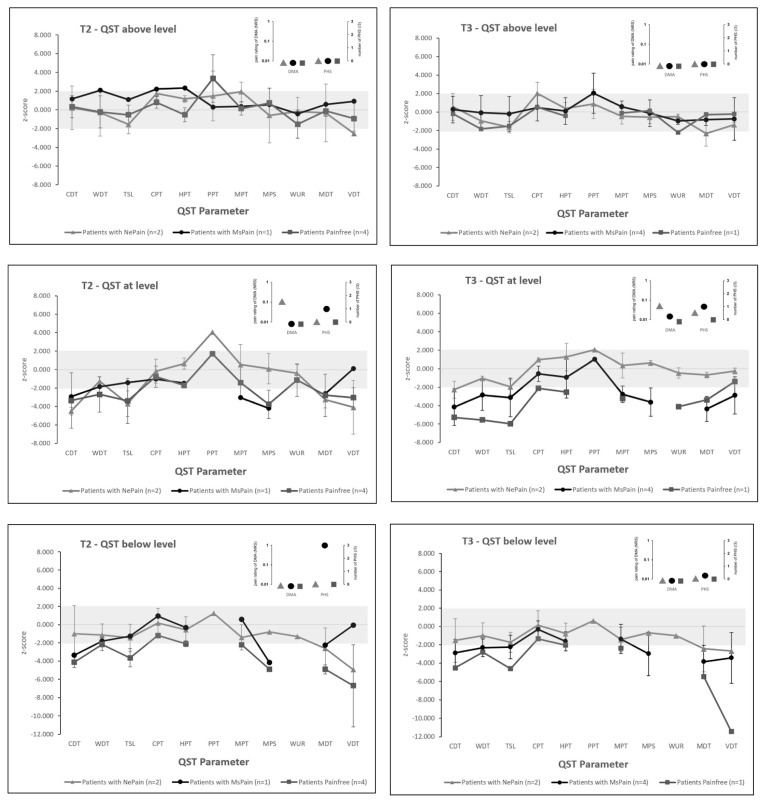
QST at timepoints T2 (**left** column) and T3 (**right** column) grouped into neuropathic pain, nociceptive pain, and no pain, measured above, at and below the level of neurological injury. Parameters are shown in z-score values, except for PHS and DMA (inset). The grey area indicates values in the normal range (−1.96 to 1.96). Z-scores below 0 indicate a loss of function in that specific QST parameter; z-scores above 0 indicate a gain of function in that specific QST parameter. Missing values are due to either deafferentation (stimuli are not felt and therefore not assessable) or spastic reactions. Timepoints: T2 (12 weeks) and T3 (24 weeks). **Abbreviations:** CDT: cold detection threshold; CPT: cold pain threshold; DMA: dynamic mechanical allodynia; HPT: heat pain threshold; MDT: mechanical detection threshold; MPS: mechanical pain sensitivity; MPT: mechanical pain threshold; NeuPain: neuropathic pain; NocPain: nociceptive pain; NRS: numeric rating scale; PHS: paradoxical heat sensation; PPT: pressure pain threshold; QST: quantitative sensory testing; TSL: thermal sensory limen; VDT: vibration detection threshold; WDT: warm detection threshold; WUR: wind-up ratio.

**Figure 4 ijms-24-11122-f004:**
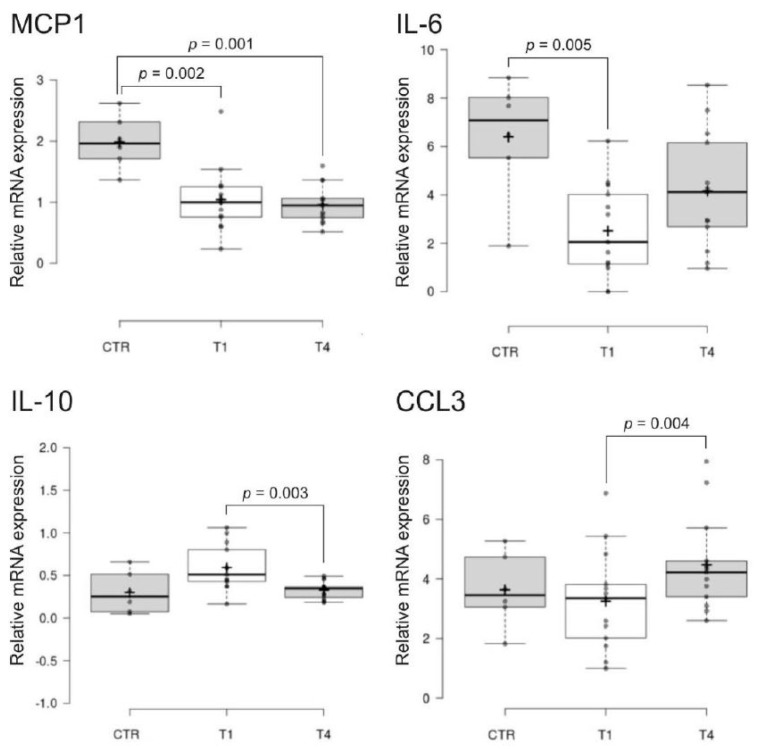
Blood biomarker gene expression analysis. Box plots represent relative mRNA expression normalized on 3 housekeeping genes (GAPDH, PPIB, and B2M). The line across the box indicates the median, the cross indicates the mean, and bubbles indicate outliers. Significance was determined at *p* < 0.05. Timepoints: T1 (4 weeks) and T4 (discharge). **Abbreviations:** CTR: healthy able-bodied individuals; CCL3: chemokine (C-C motif) ligand 3; IL: interleukin; MCP1: monocyte chemoattractant protein 1.

**Figure 5 ijms-24-11122-f005:**
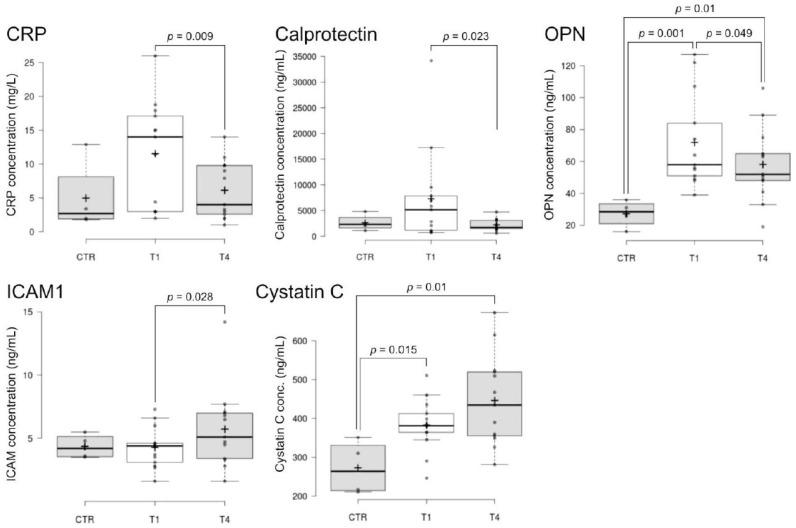
Blood biomarker flow-cytometry-based immunoassays. Box plots show the protein level of BB in serum. The line across the box indicates the median, the cross indicates the mean, and bubbles indicate outliers. Significance was determined at *p* < 0.05. Timepoints: T1 (4 weeks) and T4 (discharge). **Abbreviations:** CTR: healthy able-bodied individuals; CRP: C-reactive protein; ICAM-1: intercellular adhesion molecule 1; OPN: osteopontin.

**Table 1 ijms-24-11122-t001:** Psychosocial and pain-related questionnaires.

*Questionnaire*		T1	T2	T3	T4
**Pain interference questionnaire (0–10)**		N = 2	N = 3	N = 5	
	Daily activities	3.5 ± 0.5 (2.8–4.2)	4.0 ± 1.6 (2.2–5.8)	2.4 ± 2.6 (0.1–4.7)	=T3 ^2^
	Mood	1.0 ± 1.0 (0–2.4)	3.0 ± 3.6 (0–7)	2.8 ± 2.4 (0.7–4.9)	=T3 ^2^
	Sleep	0.0 ± 0.0 (0–0)	3.0 ± 1.6 (1.2–4.8)	1.4 ± 1.5 (0.1–2.7)	=T3 ^2^
**SCIPI (0–7)**					
	Neuropathic pain sites	3.0 ± 0.0 (3–3) N = 1	4.0 ± 0.0 (4.0–4.0) N = 3	4.5 ± 0.5 (1.2–3.8) N = 2 ^1^	=T3 ^2^
	Nociceptive pain sites	0.3 ± 0.5 (0–0.9) N = 3	2.0 ± 0.0 (2.0–2.0) N = 1	2.5 ± 1.6 (3.8–5.2) N = 6	=T3 ^2^
**Distress thermometer** **(0–10)**		N = 10	N = 12	N = 2	N = 12
		6.3 ± 2.7 (4.6–8.0)	6.2 ± 2.1 (5.0–7.4)	5.0 ± 2.8 (1.1–8.9)	4.8 ± 2.8 (3.3–6.4)
**HADS (0–21)**		N = 13	NA	N = 2	N = 12
	Depression	6.5 ± 4.2 (4.2–8.8)	NA	5.0 ± 1.4 (3.0–7.0)	6.0 ± 3.5 (4.0–8.0)
	Anxiety	5.8 ± 2.2 (4.6–7.0)	NA	5.0 ± 2.8 (1.1–8.9)	6.8 ± 3.2 (5.0–8.7)
**International SCI QoL Basic Data Set (0–10)**		N = 13	N = 11	N = 2	N = 12
Satisfaction with	General quality of life	6.1 ± 3.1 (4.4–7.7)	6.0 ± 2.8 (4.4–7.6)	6.5 ± 3.5 (1.6–11.4)	6.3 ± 1.9 (5.3–7.4)
Satisfaction with	Physical health	5.6 ± 3.1 (3.9–7.3)	5.6 ± 2.8 (4.0–7.3)	4.5 ± 3.5 (0–9.4)	5.0 ± 2.1 (3.8–6.2)
Satisfaction with	Psychological health	6.2 ± 2.8 (4.7–7.8)	5.0 ± 2.9 (3.3–6.7)	5.5 ± 2.1 (2.6–8.4)	6.2 ± 2.4 (4.8–7.5)

Data are shown as mean ± SD (95% CI); N = number of patients who filled out the questionnaires. Timepoints: T1 (4 weeks), T2 (12 weeks), T3 (24 weeks), and T4 (discharge). **Note:**
^1^ One SCIPI related to a neuropathic pain site is missing. ^2^ T4 = T3 for all patients except one, who reported no pain at T4. **Abbreviations:** HADS: Hospital Anxiety and Depression Scale; QoL: quality of life; SCIPI: Spinal Cord Injury Pain Instrument.

## Data Availability

The data supporting the findings of this study are available within the article and Appendix A.

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
