# Peer review of "Assessing the Feasibility of a Multimodal Approach to Pain Evaluation in Early Stages after Spinal Cord Injury"

_ijms, 2023, doi:10.3390/ijms241311122_

Round 1

Reviewer 1 Report

The article entitled "Assessing the feasibility of a multimodal approach to pain evaluation in early-stages after spinal cord injury" proposes to evaluate spinal cord injured patients in the initial 6-month period following trauma. The evaluation is clinical, electrophysiological and biological. The assessment is also carried out at the end of hospitalization, the average length of which is not easy to find in the article.  

The aim of this multimodal approach seems to be essentially in the field of research, but this article is not structured in this sense, even though approval from an ethics committee has been obtained and a consent form has been signed by the patients. The problem of feasibility is not in itself a research issue. In general, the number of patients to be studied is determined either by an "a priori" statistical study or by analogy with other studies already published. 

I suggest that authors follow the classic plan of a research article, with a paragraph dedicated to the study population (which is not usually part of the results), a paragraph dedicated to the evaluation methods (with at least one reference per technique) and an agenda clearly mentioning the timing of the evaluations. T1, T2, T3 and T4 must be defined in the text and, if necessary, in the abstract. These different paragraphs reappear in the discussion section, which is unusual. 

The number of patients studied is unclear. Looking at Figure 1, one wonders whether 7 or 13 patients are involved, while line 118 lists 14 patients. 

In my opinion, there are too many figures and tables. Table 2, for example, could be simplified to report only interesting or significant data.  

At no point is the practical value of this research discussed. I suggest that the article be restructured, perhaps around the 5 patients who have undergone the entire protocol, and that it be opened up to the therapeutic prospects that may arise from such a protocol. As far as I'm concerned, feasibility is just one element to be discussed. 

Author Response

We thank the Reviewer 1 for considering our manuscript:

''Assessing the Feasibility of a Multimodal Approach to Pain Evaluation in Early-Stages After Spinal Cord Injury''

We addressed concerns, requests and suggestions as follows.

Comments and Suggestions for Authors:

The article entitled "Assessing the feasibility of a multimodal approach to pain evaluation in early-stages after spinal cord injury" proposes to evaluate spinal cord injured patients in the initial 6-month period following trauma. The evaluation is clinical, electrophysiological and biological. The assessment is also carried out at the end of hospitalization, the average length of which is not easy to find in the article.  

Thank you for your valuable feedback and suggestions. We appreciate your attention to detail regarding the average length of hospitalization, and we apologize for the oversight in not explicitly mentioning this information in the manuscript. We have now included the average length of hospitalization, which was found to be 27±8.8 weeks after the injury, in the section ''Study population" [line 91]. 

The aim of this multimodal approach seems to be essentially in the field of research, but this article is not structured in this sense, even though approval from an ethics committee has been obtained and a consent form has been signed by the patients. The problem of feasibility is not in itself a research issue. In general, the number of patients to be studied is determined either by an "a priori" statistical study or by analogy with other studies already published. 

We appreciate your comment and the opportunity to clarify the research focus and address the concern regarding the structure of the article. While it is true that our study assesses the feasibility of a multimodal pain evaluation protocol, it is important to emphasize that this assessment is conducted as a nested project within the framework of the SwiSCI Cohort Study.

To provide more insight into the research design and statistical considerations, we have now included additional information in the Materials and Methods section, specifically in the paragraph titled "Patient Recruitment" [line 379].  In this section, we have included details from the SwiSCI nested project study proposal, which envisioned the recruitment of 150 spinal cord injured (SCI) patients over a period of 3 years. Based on an estimated effect size of Cohen's d = 0.5 (in case of any pain treatment), a total sample of 118 participants was determined to provide 80% power to reject the null hypothesis with an α level of 0.05.

However, we encountered challenges in recruitment during the study period of 15 months, resulting in a very low recruitment rate. Out of the 117 SCI approached patients, only 8 consented to participate in the neurological and pain evaluation within the SwiSCI nested project. We acknowledge that this recruitment problem was a limitation of our study and may affect the generalizability of the findings, however conducting the pilot trial had the aim to investigate the feasibility of recruitment, so this is a valid feasibility trial result. We have included this information in the ''Study population'' section and in the Discussion.

I suggest that authors follow the classic plan of a research article, with a paragraph dedicated to the study population (which is not usually part of the results), a paragraph dedicated to the evaluation methods (with at least one reference per technique) and an agenda clearly mentioning the timing of the evaluations. T1, T2, T3 and T4 must be defined in the text and, if necessary, in the abstract. These different paragraphs reappear in the discussion section, which is unusual. 

Thank you for your suggestion regarding the structure of the research article. We agree that following a classic plan will enhance the organization and readability of our manuscript. We have taken your advice into consideration and made the necessary revisions to address these concerns.

  1. Study Population: We have now included a dedicated section after the Introduction, titled "Study Population", which provides a detailed description of the participants and the variability in the number of participants undergoing neurological and pain evaluation, blood biomarker analysis, and psycho-social factor assessments. This section offers a comprehensive overview of the study population, addressing your suggestion to include a paragraph specifically dedicated to this aspect.
  2. Evaluation Methods: We apologize for the confusion in our previous response. In the Materials and Methods section, we have indeed referenced the evaluation methods [line 373], providing a description of the clinical workup (CW), quantitative sensory testing (QST), psycho-social factors (PSF), and molecular blood biomarkers (BB). To further improve clarity, we have included at least one reference per evaluation technique, highlighting the relevant literature supporting their usage.
  3. Timing of Evaluations: We have incorporated a clear agenda in the Study Population section, outlining the timing of evaluations at 4 (T1), 12 (T2), 24 (T3) weeks post-injury, and at discharge (T4). This information helps readers understand the temporal framework of the study and the intervals at which assessments were conducted. Additionally, we have addressed your suggestion to define the abbreviations T1, T2, T3, and T4 in the Abstract for better clarity and comprehension.
  4. Discussion Section: We have carefully reviewed the organization of the Discussion section. While it is unusual for the paragraphs dedicated to the study population, evaluation methods, and timing of evaluations to reappear in the Discussion section, we believe that it provides a cohesive and comprehensive overview of our research findings and their implications. However, we have made certain adjustments to ensure a smooth transition between sections and to avoid redundancy.

The number of patients studied is unclear. Looking at Figure 1, one wonders whether 7 or 13 patients are involved, while line 118 lists 14 patients. 

We have taken your feedback into account and made the necessary improvements to address this issue.

Figure 1: We have revised Figure 1 to provide clearer information regarding the number of patients investigated in each aspect of the study. The revised figure now explicitly states the number of patients involved in neurological and pain evaluation, blood biomarkers, and psycho-social factors. This modification ensures that readers can easily understand the patient distribution for different assessments.

Study Population section: We have also revised the text in the Study Population section to provide a clear and accurate description of the number of patients studied in each aspect. We have explicitly stated that 7 patients were investigated for neurological and pain evaluation, 13 patients for blood biomarkers, and 14 patients for psycho-social factors. By including this clarification, we aim to alleviate any confusion regarding the patient numbers.

In my opinion, there are too many figures and tables. Table 2, for example, could be simplified to report only interesting or significant data.  

We appreciate your suggestion to simplify Table 2 and include only the most relevant data.

Table 2: We have taken your suggestion into consideration and decided to move the content of Table 2 to the Supplementary Materials as Table S8. This allows us to streamline the presentation of data and focus on the most interesting and significant findings in the main manuscript. The relevant and significant data from Table 2 are represented in Figures 4 and 5, which provide a clear visual representation of the results.

Table 3: We have reevaluated the need for Table 3 in the manuscript, considering your comment that feasibility aspects were already described in the text. We have decided to remove Table 3 from the manuscript.  

At no point is the practical value of this research discussed. I suggest that the article be restructured, perhaps around the 5 patients who have undergone the entire protocol, and that it be opened up to the therapeutic prospects that may arise from such a protocol. As far as I'm concerned, feasibility is just one element to be discussed.

Thank you for your insightful comment regarding the practical value of our research and the necessity of discussing its therapeutic prospects. We agree that highlighting this aspect of the research will augment its relevance and translational value.

The feasibility trial we presented is indeed a subset of the larger SwiSCI Cohort study, which routinely collects a wide variety of functional variables and bio-samples during initial SCI rehabilitation. Our aim was to examine the potential for integrating a bio-psycho-social pain evaluation protocol within this longitudinal study, a task we acknowledge to be rather complex due to the consent process for the study participants.

The participants were required to provide informed consent for the cohort study itself, as well as separate consents for blood sample donation and QST pain evaluation. This multiplicity of consents presented us with uncertainty over potential recruitment bottlenecks, leading us to hypothesize that participants might be less willing to provide consent for blood sample donation. Contrary to our initial expectation, however, it was the QST pain evaluation consent that critically affected recruitment.

While we acknowledge the reviewer's suggestion to restructure the article around the 5 patients who consented to all aspects of the study, we feel it important not to overlook the potential implications of the consent challenge in our research. By providing a comprehensive account of our recruitment and consent experiences, we believe that we can offer valuable insights for researchers designing similar studies in the future. The Flow Diagram in Figure 1 serves as a resource for identifying potential challenges in participant recruitment for future multimodal pain studies.

However, recognizing the validity of the reviewer's suggestion, we have included an additional analysis focused on the 5 patients who consented to the entire protocol. This analysis, which discusses the implications of our findings for the development of therapeutic interventions, is included in the "Blood biomarkers" paragraph of the Results [line 232].

Our intent with this approach is twofold. Firstly, we aim to satisfy the reviewer's request for an exploration of the practical value and therapeutic prospects of our research. Secondly, we are committed to preserving a comprehensive account of our methodology and experiences to facilitate future research in this field.

We have also added a summary of this topic to the discussion section, which has been thoroughly revised.

Reviewer 2 Report

  This manuscript deals with a very important but under-evaluated issue of pain evaluation in spinal cord injury (SCI) patients. The authors conclude that their study shows that multimodal pain evaluation is feasible in the early stages of SCI, but after reviewing the manuscript this reviewer actually feels that this study demonstrates the difficulty of sufficiently evaluating SCI patients in the acute phase.

  There are numerous issues that the authors need to address.

  This study is introduced as a nested project within the Swiss Spinal Cord Injury (SwiSCI) cohort study, but the authors do not give any information about the SwiSCI study. This reviewer asks the authors to provide basic information about the inclusion criteria of the SwiSCI study so that readers understand the nature of the patients studied.

  The greatest limitation of this study is the extremely limited number of participants and the spotty collection of data, which should make the authors wary of drawing any conclusions. The authors try to evaluate the results, but this reviewer has strong reservations about making any generalized assessments from such a limited subject pool. Please reevaluate the scientific basis of the interpretations made of the results.

  Another point that makes evaluation of the results difficult is the high variability of the nature of the SCI in the included patients. There are only seven patients included in the pain trajectory evaluation, but there are cervical, thoracic, and epiconus level patients as well as complete AIS A injuries to incomplete AIS D injuries. These are very different injuries that should not be evaluated together. This also extends to the blood biomarker studies as well. With such a variability in injury level and severity, how can the authors draw any conclusions from any trend seen in the results from the 13 patients included in the study for blood biomarkers? Unlike the information provided for the seven patients included in the pain trajectory evaluation, no information is provided about the SCI level or severity of the remaining six patients. Another small point is that the control patients show an elevated CRP levels; any reason for this inflammation?

  Unless these points can be resolved and put together in a coherent form, this reviewer does not feel that the results of this study provides any new information to the readers of IJMS.

There are some minor grammatical errors and typos that can be weeded out during the editorial process.

Author Response

We thank the Reviewer 2 for considering our manuscript:

''Assessing the Feasibility of a Multimodal Approach to Pain Evaluation in Early-Stages After Spinal Cord Injury''

We addressed concerns, requests and suggestions as follows.

Comments and Suggestions for Authors:

  This manuscript deals with a very important but under-evaluated issue of pain evaluation in spinal cord injury (SCI) patients. The authors conclude that their study shows that multimodal pain evaluation is feasible in the early stages of SCI, but after reviewing the manuscript this reviewer actually feels that this study demonstrates the difficulty of sufficiently evaluating SCI patients in the acute phase.

We value your interpretation of our work, and we agree that our study indeed underscores the inherent challenges in adequately assessing pain in SCI patients during the acute phase.

While these challenges were palpable, our intention was to illuminate the possible course for such evaluation and to identify the limiting factors. Our study indeed elucidates these limitations, including recruitment and adherence issues, protocol compliance, and variability in the patient's response. This kind of insight is invaluable as it paves the way to improving future protocols and methodologies, thereby augmenting the quality of pain evaluation in the early stages post-SCI.

We are mindful of the complexity surrounding these evaluations but are also cautiously optimistic about the future. Drawing upon the lessons learned from this research, we believe that an improved protocol that accounts for these limitations can be developed. We foresee that by actively managing these identified limiting factors, we can enhance recruitment rates and ultimately improve the feasibility of conducting such evaluations.

We have also added a summary of this topic to the discussion section.

  There are numerous issues that the authors need to address.

This study is introduced as a nested project within the Swiss Spinal Cord Injury (SwiSCI) cohort study, but the authors do not give any information about the SwiSCI study. This reviewer asks the authors to provide basic information about the inclusion criteria of the SwiSCI study so that readers understand the nature of the patients studied.

We have supplemented our manuscript with additional information about the SwiSCI study in the ''Study population'' and in the paragraph ''Study design'' in ''Materials and Methods'' section.

The SwiSCI is a prospective longitudinal cohort study initiated in 2011, involving individuals with spinal cord injuries from across Switzerland. The study encompasses diverse aetiologies of spinal cord injury, including traumatic and non-traumatic causes, and includes all lesion severities, lesion levels, and stages after injury, thus providing a representative sample of the SCI population in Switzerland. To be included in the study, participants must have a confirmed diagnosis of spinal cord injury, be at least 16 years old, reside in Switzerland, and give informed consent to participate. Participants are excluded if they have a congenital disorder leading to comparable symptoms of SCI but without a primary lesion in the spinal cord (such as spina bifida or polio).

For more detailed information about the SwiSCI study, including its design, participant characteristics, response rates, and non-response, we direct readers to the original study publications:

  1. Post, M.W.; Brinkhof, M.W.; von Elm, E.; Boldt, C.; Brach, M.; Fekete, C.; Eriks-Hoogland, I.; Curt, A.; Stucki, G.; Swi, S.C.I.s.g. Design of the Swiss Spinal Cord Injury Cohort Study. Am J Phys Med Rehabil 2011, 90, S5-16, doi:10.1097/PHM.0b013e318230fd41.

  1. Fekete, C.; Gurtner, B.; Kunz, S.; Gemperli, A.; Gmunder, H.P.; Hund-Georgiadis, M.; Jordan, X.; Schubert, M.; Stoyanov, J.; Stucki, G. Inception cohort of the Swiss Spinal Cord Injury Cohort Study (SwiSCI): Design, participant characteristics, response rates and non-response. J Rehabil Med 2021, 53, jrm00159, doi:10.2340/16501977-2795.

  The greatest limitation of this study is the extremely limited number of participants and the spotty collection of data, which should make the authors wary of drawing any conclusions. The authors try to evaluate the results, but this reviewer has strong reservations about making any generalized assessments from such a limited subject pool. Please reevaluate the scientific basis of the interpretations made of the results.

We agree with your comments about the small participant pool and occasional gaps in data collection. These issues indeed present challenges when it comes to drawing robust conclusions. However, we believe that these very limitations underscore the significant need for the study at hand and also highlight the areas that demand improvement in future research.

Our primary objective was to explore the feasibility of a multimodal approach to pain assessment during the early stages following spinal cord injury. While we attempt to interpret and discuss the results of the assessments, we recognize and appreciate your concern about the possibility of overgeneralizing from a limited dataset. Consequently, we have revised the manuscript to emphasize the preliminary nature of these results and to indicate clearly that our interpretations are merely exploratory.

We have strived to be cautious in our language, ensuring that we do not imply direct causality between observed phenomena and neuropathic pain. Instead, we present our findings as a collection of potential indicators that may warrant further investigation in more extensive studies.

  Another point that makes evaluation of the results difficult is the high variability of the nature of the SCI in the included patients. There are only seven patients included in the pain trajectory evaluation, but there are cervical, thoracic, and epiconus level patients as well as complete AIS A injuries to incomplete AIS D injuries. These are very different injuries that should not be evaluated together. This also extends to the blood biomarker studies as well. With such a variability in injury level and severity, how can the authors draw any conclusions from any trend seen in the results from the 13 patients included in the study for blood biomarkers?

We agree with your assertion that the diversity in injury type and severity could confound the interpretation of both the pain trajectories and the blood biomarker studies. This issue is one that confronts much of SCI research due to the complex and multifaceted nature of these injuries. However, we believe that this heterogeneity also reflects the reality of spinal cord injuries in a clinical setting, where the nature and severity of injuries are seldom uniform. Our primary aim was to assess the feasibility of our multimodal approach and to provide a preliminary exploration of potential trends and associations, which we hope will inspire more comprehensive studies.

We recognize, as you rightly point out, that grouping disparate injuries together could distort the findings. To manage this concern, it has been proposed to apply a statistical approach known as latent class growth mixture modeling (ref. Jung T, Wickrama KAS: An introduction to latent class growth analysis and growth mixture modeling. Social and Personality Psychology Compass. 2008;2). This methodology allows to identify homogeneous subgroups within our heterogeneous sample, delineating meaningful classes or groups of individuals based on variations in their pain trajectories over time. This approach should provide a more nuanced understanding of our findings while accounting for the heterogeneity of our sample. Due to the feasibility nature of our study and the low recruitment rate this method hasn't been used in our case.

Unlike the information provided for the seven patients included in the pain trajectory evaluation, no information is provided about the SCI level or severity of the remaining six patients.

Clinical data of patients analyzed for blood biomarkers are showed in Supplementary Table S7 with information about the level of injury and the ASIA Impairment scale.

Another small point is that the control patients show an elevated CRP levels; any reason for this inflammation?

Thank you for your suggestion regarding healthy values of CRP.

A CRP level lower than 3 mg/L is considered normal and 3-10 mg/L is considered normal or minor elevation (C Reactive Protein - StatPearls - NCBI Bookshelf (nih.gov)).

In our study, the CRP value in healthy control samples is on average 5.0 +/- 5.3 mg/L, because of one healthy donor with high CRP value of 12.9 mg/L. Excluding this donor, the CRP average value is 2.4 +/- 0.8 mg/L (normal level). We do not have additional clinical information about the healthy donor to explain the high CRP value. The new calculation of the healthy CRP value was included in the ''Blood Biomarkers'' paragraph of Results section [line 228].

  Unless these points can be resolved and put together in a coherent form, this reviewer does not feel that the results of this study provides any new information to the readers of IJMS.

Comments on the Quality of English Language

There are some minor grammatical errors and typos that can be weeded out during the editorial process.

We thank the editorial team for this proposal.

Round 2

Reviewer 1 Report

The authors have responded perfectly to my comments. I have no further comments to make.

Reviewer 2 Report

The comments by the authors and the revisions to the manuscript address some of the issues raised by this reviewer. Although this reviewer still has some reservations about the generalizations made in this manuscript, I have decided that the information provided by this manuscript is beneficial for clinicians and researchers involved in the care of spinal cord injury patients. 

I believe there are some typos remaining.